# Recent Advances in MOF-based Nanocatalysts for Photo-Promoted CO$_2$ Reduction Applications

**Chang Liu [1], Wenzhi Wang [2], Bin Liu [2], Jing Qiao [1], Longfei Lv [1], Xueping Gao [1,*], Xue Zhang [1], Dongmei Xu [3], Wei Liu [3], Jiurong Liu [1,*], Yanyan Jiang [1], Zhou Wang [1], Lili Wu [1] and Fenglong Wang [1,*]**

1   School of Materials Science and Engineering, Shandong University, Jinan 250061, China
2   School of Materials Science and Engineering, University of Jinan, Jinan 250022, China
3   State Key Laboratory of Crystal Materials, Shandong University, Jinan 250061, China
*   Correspondence: xpgao@sdu.edu.cn (X.G.); jrliu@sdu.edu.cn (J.L.); fenglong.wang@sdu.edu.cn (F.W.);
    Tel.: +86-531-883-99579 (F.W.)

**Abstract:** The conversion of CO$_2$ to valuable substances (methane, methanol, formic acid, etc.) by photocatalytic reduction has important significance for both the sustainable energy supply and clean environment technologies. This review systematically summarized recent progress in this field and pointed out the current challenges of photocatalytic CO$_2$ reduction while using metal-organic frameworks (MOFs)-based materials. Firstly, we described the unique advantages of MOFs based materials for photocatalytic reduction of CO$_2$ and its capacity to solve the existing problems. Subsequently, the latest research progress in photocatalytic CO$_2$ reduction has been documented in detail. The catalytic reaction process, conversion efficiency, as well as the product selectivity of photocatalytic CO$_2$ reduction while using MOFs based materials are thoroughly discussed. Specifically, in this review paper, we provide the catalytic mechanism of CO$_2$ reduction with the aid of electronic structure investigations. Finally, the future development trend and prospect of photocatalytic CO$_2$ reduction are anticipated.

**Keywords:** Metal-organic frameworks (MOFs); photocatalysis; carbon dioxide reduction; renewable energy

---

## 1. Introduction

Energy shortages and environment issues are global problems and challenges that are faced by human beings today [1–6]. The development of renewable energy technologies to reduce the pollutants emission has become an important research topic to maintain the sustainable development of our planet [7–12]. Artificial photosynthesis is an ideal way to effectively solve the energy and environmental problems by decomposing water to produce hydrogen or reducing CO$_2$ to high value-added chemicals or fuels [13–16]. Accordingly, searching for highly efficient materials that can convert solar energy and store it in chemicals is desired. Metal-organic frameworks (MOFs), which are known as coordination porous polymer, is a class of crystalline porous materials constructed by the coordination bond between metal ions or metal cluster nodes [17–21]. These materials have been widely used in gas separation/storage, catalysis, sensing, proton conductors, and drug delivery because of their structural diversity, design/modification, and ultra-high specific surface areas [22–25]. Based on the previous results, it is proven that the multifunctional organic ligands in the MOFs structure can play the role of "light capture antenna" [26,27]. It can effectively accept photons, generate band gap transition, and transfer electrons to metal center units. Thus, MOFs are usually used as efficient photocatalysts [28–30]. When comparing to other photocatalytic materials, MOFs exhibit big specific surface area, high porosity, and supervised capturing capability of CO$_2$ molecules, which endows

them with great application prospect in the field of photocatalysis for $CO_2$ reduction. In recent years, MOFs and their composite materials are widely used in water decomposition, hydrogen production, $CO_2$ reduction, and photocatalytic organic conversion [31].

Yaghi group first proposed the concept of the metal-organic frameworks in 1995, and MOFs materials were then intensively explored as new functional materials [32]. In 1997, Kitagawa group reported a three-dimensional MOF material and found its ability to adsorb gas at room temperature [33]. After that, two landmark cases of MOFs, MOF-5 and HKUST-1, were reported by Yaghi group and Williams group in 1999 [34–36]. Among them, MOF-5 is a three-dimensional skeleton that formed by coordination of $Zn_4O(CO_2)_6$ clusters and terephthalic acid ligands. Through the gas adsorption experiments, the authors found that MOFs-5 showed high specific surface area, large pore size, and a certain adsorption capacity for hydrogen. HKUST-1, as reported, is a three-dimensional skeleton that is formed by the coordination of $Cu_2(CO_2)_4$ clusters with benzotriformic acid ligands [37]. The authors found that HUKST-1 with unsaturated ligand sites can be obtained by heating water molecules that can be removed and coordinated on metal clusters [38]. Jinhee et al. report the the OCS-activation ability of chloromethanes to remove precoordinated solvent molecules from open coordination sites (OCSs) in MOFs [39]. A water molecule in HKUST-1 can easily access open metal site (OMS)with high coordination strength due to the specific coordination geometry around $Cu^{2+}$ [40]. In particular, MOFs with OCSs have potential applications in chemical separation, molecular sorption, catalysis, ionic conduction, and sensing areas [39,41]. Since these two MOF structures were reported, the synthesis and potential applications of MOFs in gas separation, storage, catalysis, sensing, drug transportation, and so on have become hot research topics [42,43].

MOFs are extensively studied for the capture and conversion of $CO_2$ due to their high porosity and strong interaction with $CO_2$ molecules. At present, some MOFs have already been explored for their high catalytic performance in the field of photocatalysis for $CO_2$ reduction [44]. As photocatalysts, MOFs exhibit the following advantages. Firstly, the high specific surface area of MOFs is helpful for the gas reactants adsorption around the active site. This is beneficial to the molecule activation and catalytic transformation in the subsequent process [45,46]. Secondly, the metal-oxygen units in MOFs exhibit semiconductor-like structure due to the existence of organic ligands. MOFs with larger energy than the band gap can be excited by photons to create electron and hole pairs [47,48]. Through selectively choosing different organic ligands and metal centers, one can improve the absorption and utilization efficiency of sunlight via MOFs as light absorbing agents [49]. Besides, the separation and transfer of electrons can be promoted by changing the crystal structure, thereby which thereby inhibits the recombination of photo-induced electrons and holes [50]. In addition, MOFs, as heterogeneous catalysts, can be easily separated and recycled from the reaction system, which is beneficial for prolonging the service life of the catalyst and avoiding any pollution to the environment [51–53].

In this paper, the advances of MOFs materials for photocatalytic $CO_2$ reduction is systematically reviewed. This review paper starts from the research background why $CO_2$ reduction is important, and the mechanism studies on the photocatalytic $CO_2$ reduction process were then summarized. After that, the research progress of photocatalytic $CO_2$ reduction using MOFs were reviewed, followed by the summary of the applications of MOFs-based composite materials for photocatalytic reduction of $CO_2$. Finally, the current challenges and future development trend of MOFs-based materials for photocatalytic $CO_2$ reduction are anticipated.

## 2. Necessity

A large amount of fossil fuels has been combusted since the eighteenth century, so that the atmospheric $CO_2$ concentration increased gradually. According to the data of the National Oceanic and Atmospheric Administration (NOOA), the $CO_2$ concentration has exceeded 400 ppm in May, 2013, and it reached 402 ppm in May 2014 [54]. It is believed that the atmospheric $CO_2$ concentration will exceed 550 ppm at the end of this century [55,56]. The sudden increase of $CO_2$ concentration in the

atmosphere can be attributed to the over-use of the fossil fuels. Currently, more than 80% of the global energy supply origins from fossil fuels, which generates a large amount of $CO_2$ in the atmosphere.

A hundred years ago, Arrhenius suggested that $CO_2$ emissions from the burning of fossil raw materials would lead to an increase in global temperatures [57,58]. Today, $CO_2$ has been widely accepted as the chief culprit causing global warming, climate upheaval, and many other environmental problems. Various environmental problems will become much sharper if there are no effective measures are taken to curb $CO_2$ emissions [59]. When the atmospheric $CO_2$ content rises to 450 ppm, the accompanying increase in global temperature will seriously aggravate the cessation of the hot salt circulation, and environmental problems, like melting of glaciers, will take place [60].

In the 21st century, in addition to serious environmental problems, the energy crisis is also a global issue affecting human society. In 2008, the total global energy consumption was 132,000 megawatts. According to the U.S. [61] Energy Information Administration, this number will continuously grow, and the total energy consumption in 2040 is expected to be twice of that in 2020. Although the over exploitation and use of fossil energy has caused global warming and energy crisis, we can still find some opportunities and challenges to debate these issues [62]. For example, while using a suitable method to convert $CO_2$ into energy materials or valuable industrial raw materials is a promising solution to close the carbon loop, and can alleviate the dependence of human beings on fossil energy and solve environmental problems that are caused by $CO_2$ emissions [63,64].

## 3. Mechanisms

Photocatalytic $CO_2$ reduction involves three basic processes. Under light irradiation, the electron-hole pairs could be generated in semiconductor materials upon the absorption of photons with larger energy than the forbidden band gap [65]. Subsequently, the photoexcited electron-hole pairs separate and migrate to the active sites on the surface of the semiconductor. In this process, it is necessary to reduce the bulk phase and surface recombination of photogenerated electron-hole pairs. This is the major factor limiting the efficiency of photocatalytic reduction of $CO_2$ [66,67]. After that, oxidation and reduction reactions occur on the surface of the semiconductor. At this time, electrons with strong enough reducing ability can reduce $CO_2$ molecules into hydrocarbons, such as CO, $CH_4$, and $CH_3OH$, and holes with oxidizing ability oxidize $H_2O$ molecules to release $O_2$, $O^{2-}$, and other substances [68]. The conversion efficiency of photocatalytic $CO_2$ reduction depends on the capacity of the light-trapping ability of the semiconductor material, the efficiency of photo-generated carrier generation and separation, and the thermodynamic equilibrium of the surface catalytic reactions. From the kinetic point of view, the effective absorption of light, the efficient separation and migration of photo-generated electron-hole pairs, and the sufficient reactive sites on the catalyst surface are an important prerequisite for the high-efficiency photocatalytic conversion of $CO_2$ while using semiconductor materials [69].

The detailed mechanisms for photocatalytic $CO_2$ reduction process have not been discovered so far. However, mechanism studies in recent years provide valuable information to unravel this process [70]. At present, it is commonly accepted that photocatalytic $CO_2$ reduction is a multi-electron reduction process, as described in the Equations (2)–(8). It can be seen that the reaction process is accompanied by some unstable substances, namely intermediates. The corresponding products are different due to the specific reaction route and the number of electrons obtained during the reaction [71,72]. According to the number of electrons that were obtained by C atom, the products can be carbon monoxide, methane, formic acid, methanol, etc. [73]. In some special reaction system, some multi-carbon compounds such as ethane, acetic acid, and other compounds can also be obtained. From the perspective of Gibbs free energy, photocatalytic reduction of $CO_2$ is an uphill reaction, that is $\Delta G > 0$. If the reaction proceeds, a large amount of energy injection (such as incident photons) is required.

Reaction Eredox/ (V vs NHE,PH=7)

$$CO_2 + e^- \rightarrow CO^- \qquad -1.90 \tag{1}$$

$$CO_2 + H^+ + 2e^- \rightarrow HCO_2^- \qquad -0.49 \tag{2}$$

$$CO_2 + 2H^+ + 2e^- \rightarrow CO + H_2O \qquad -0.53 \tag{3}$$

$$CO_2 + 4H^+ + 4e^- \rightarrow HCHO + H_2O \qquad -0.48 \tag{4}$$

$$CO_2 + 6H^+ + 6e^- \rightarrow CH_3OH + H_2O \qquad -0.38 \tag{5}$$

$$CO_2 + 8H^+ + 8e^- \rightarrow CH_4 + 2H_2O \qquad -0.24 \tag{6}$$

$$2H^+ + 2e^- \rightarrow H_2 \qquad -0.41 \tag{7}$$

$$H_2O \rightarrow 0.5\,O_2 + 2H^+ + 2e^- \qquad 0.82 \tag{8}$$

Hendon et al. [74] elucidated the electronic structure of MIL-125 with aminated linkers through a combination of synthesis and computation. They also discussed the band gap modification of MIL-125, a $TiO_2$/1,4-benzenedicarboxylate (bdc) MOF, and the possible mechanism for the photocatalytic $CO_2$ reduction was proposed (Figure 1).

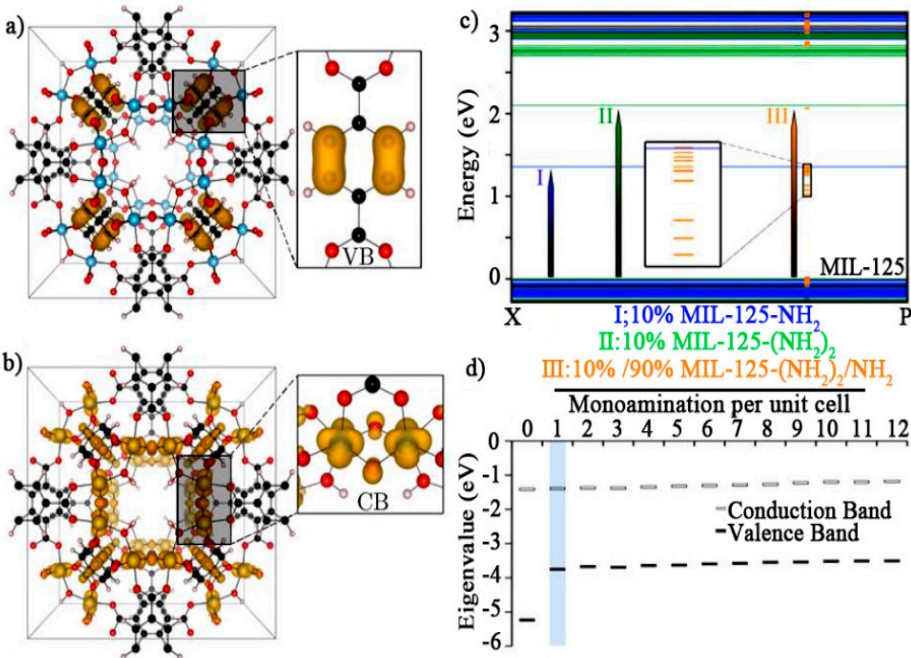

**Figure 1.** (**a**) the valence band is composed of the bdc C 2p orbitals (shown on the right), making these favorable for linker-based band gap modifications; (**b**) the conduction band is composed ofO 2p orbitals and Ti 3d orbitals (shown on the right). (**c**) PBEsol band structures for synthetic MIL-125 (black), 10%-MIL-125-NH2 (blue), 10%-MIL-125-(NH2)2/90%-MIL-125-NH2 (orange) and the theoretical 10%-MIL-125-(NH2)2 (green). (**d**) HSE06-calculated VB and CB energies of MIL-125-NH2 models containing increasing numbers of bdc-NH2 linkers [i.e. 0 (MIL-125) to 12 (100%-MIL-125-NH2)] per unit cell. MOFs materials for photocatalytic $CO_2$ reduction. Reprinted from ref. 74 with permission by the American Chemical Society.

Photocatalytic $CO_2$ reduction using MOFs-based materials as catalysts has drawn dramatic research interests in recent years. It is easy to design MOFs materials with accessible metal sites, specific hetero-atoms, and the ordered structure of functionalized organic ligands. This can effectively improve the efficiency of electron-hole separation and the photocatalytic performance. Porosity can make MOFs

expose more active sites and channels for reactant adsorption. This can improve the charge transfer efficiency as well as improve its utilization efficiency of solar energy while inhibiting the recombination of the photo-induced electron-hole pairs in the bulk phase. Based on the above merits, people try to use different MOFs for photocatalytic $CO_2$ reduction. In the following text, we will introduce three typical MOFs for photocatalytic $CO_2$ reduction and their catalytic performances. New insights for the dominating factors on activity and selectivity of product will also be discussed. Table 1 summarizes the research progress of several typical MOF materials for photocatalytic $CO_2$ reduction in recent years.

**Table 1.** the research progress of several typical metal-organic frameworks (MOF) materials for photocatalytic $CO_2$ reduction.

| Sample | Light Source Conditions | Product | Productivity | Ref. |
|---|---|---|---|---|
| $Zr_6O_4(OH)_4(bpdc)_6$ | Visible light | CO | - | 75 |
| MIL-101(Fe) | Visible light | $HCOO^-$ | 7.375μmol/h | 76 |
| PCN-222 | Visible light | $HCOO^-$ | 3.12μmol/h | 77 |
| NNU-28 | Visible light | dicarboxylic acid | 183.3μmol/h | 78 |
| $Zr_6O_4(OH)_4(L)•6DMF$ | Visible light | $HCOO^-$ | 96.2μmol/ h | 79 |
| $NH_2$-Uio-66(Zr) | Visible light | $HCOO^-$ | 1.32μmol/h | 80 |
| Ag-Ren-MOF | Visible light | CO | - | 81 |
| UiO-66-CAT | Visible light | HCOOH | 9μmo/h | 82 |
| MOF-525-Co | Visible light | CO | 36.67μmol/h | 83 |
| $Cd_{0.2}Zn_{0.8}S@UiO-66-NH_2$ | Visible light | $CH_3OH$ | - | 84 |
| Co-ZIF-9 | Visible light | CO | 28.54μmol/h | 85 |
| ZIF-67 | Visible light | CO | 3.89μmol/h | 86 |
| Ag@Co-ZIF-9 | Visible light | CO | 28.4μmol/h | 87 |
| Zn-MOF nanoliths | Visible light | CO | - | 88 |
| $Zn_2GeO_4$/Mg-MOF-74 | Visible light | CO | 1.43μmol/h | 89 |
| $TiO_2$-ZIF-8 | Visible light | MeOH | 1.21μmol/h | 90 |
| Zn/PMOF | Visible light | $CH_4$ | 10.43μmol/h | 91 |
| Co-ZIF-9/$TiO_2$ | Visible light | $CH_4$ | - | 92 |
| Cu-$TiO_2$/ZIF-8 | UV-light | CO | - | 93 |
| $CsPbBr_3@ZIFs$ | Visible light | CO | 29.630μmol/h | 94 |
| $Ti_8O_8(OH)_4(bdc)_6$(MIL-125(Ti)) | 365nm UV-light | $HCOO^-$ | 0.814μmol/ h | 95 |

*3.1. Zr MOFs*

In 2011, Wang et al. [75] chelated metal ions (such as Ir, Re, and Ru) with 4,4-biphenyldicarboxylic acid derivatives as organic ligands to construct MOFs, and the Zr-based MOF (UiO-67) systems with different metal doping were obtained. A similar synthesis strategy has also been adopted by Wang et al. who used ligand $H_2L_4$ for photocatalytic reduction of $CO_2$ to CO [76]. The total conversion number (TON) of $CO_2$ reduction can reach 10.9. The photocatalytic activity can be improved by doping a variety of photoactive metal nanoparticles inside MOFs. Subsequently, the authors observed a significant decrease in photocatalytic activity through a series of comparison experiments, which proved that the metal nanoparticles themselves are the real active sites that are involved in the photocatalytic reaction.

In 2015, Xu et al. [77] chose Zr-MOF (PCN-222) containing porphyrin as catalysts and found that it could be used as a stable photocatalyst to reduce $CO_2$ to formate ion under visible light. It was found that PCN-222 exhibited broad-spectrum absorption properties. There existed a series of extremely long lifetime electron trap states in the material, which could inhibit the recombination of photogenerated charge carriers and improve the photoreduction efficiency of $CO_2$. In 2016, Chen et al. [78] synthesized a new microporous stable zirconium-based metal organic skeleton (NNU-28) from 4,4′-(anthracene-9,10-bis (2,1-ethynylphenyl) dicarboxylic acid, which was used to reduce $CO_2$ to formate while using triethanolamine as the sacrifice agent. Under visible light irradiation, the rate of catalytic conversion of $CO_2$ to formate ion was 183.3 μmol/h. It was found that, in the catalytic reaction, the ligands produced about 27.3% formate ions, while the metal clusters

produced about 77.7% formate ions. Under light irradiation, anthracene derivative ligands not only acted as an effective light collector, but it also sensitized $Zr_6$ oxygen clusters through the LMCT (linker-to-metal charge transfer) process. At the same time, the ligand itself can also be stimulated to form free radicals and produce photogenerated electrons. Figure 2 shows two catalytic pathways for the reduction of $CO_2$ to formate. This strategy is helpful for the design and development of MOFs materials with efficient visible light response [78].

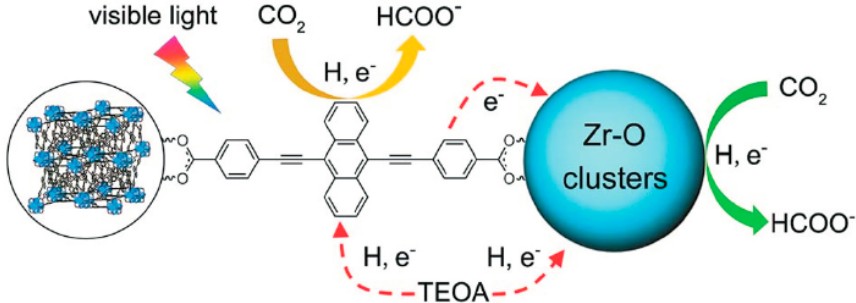

**Figure 2.** Two catalytic pathways for the reduction of $CO_2$ to formate. Reprinted from ref. 78 with permission by the Royal Society of Chemistry.

In 2018, Sun et al. [79] synthesized a porous zirconium based metal-organic framework $[(Zr_6O_4(OH)_4(L)\cdot6DMF)$ while using dicarboxyl ligands $(H_2L=2,2'$-diamino -4,4'-stilbene dicarboxylic acid, DMF) with conjugated imine function. The materials showed high chemical stability and remarkable visible light absorption properties. The average rate of $HCOO^-$ formation of MOFs is about 96.2 μmol/h.

Sun et al. [80] compared the activity of $NH_2$-UiO-66(Zr) and $NH_2$-MIL-125(Ti) for photocatalytic reduction of $CO_2$ under visible light. The results showed that the catalytic performance of $NH_2$-UiO-66(Zr) was higher than that of $NH_2$-MIL-125(Ti) under the same reaction conditions. This is ascribed to the effective transfer of photogenerated electrons from ATA to Zr-O clusters, and made Zr-O clusters efficient photocatalytic active sites. Furthermore, some ATA ligands were replaced by 2,5-diamino terephthalic acid (DTA) and the mixed ligand $NH_2$-UiO-66(Zr) was obtained. It was found that the $CO_2$ conversion of mixed $NH_2$-UiO-66(Zr) was 50% higher than that of pure $NH_2$-UiO-66(Zr). This may be because the mixed $NH_2$-UiO-66(Zr) showed strong photoabsorption capacity and large $CO_2$ adsorption capacity, so its photocatalytic activity is obviously improved.

Choi et al. [81] reported the synthesis of composited catalysts by covalently binding $Re^I(CO)_3(bpydc)Cl$(as Re TC) to UiO-67 to $Re_n$-MOFs (n is the density of Re TC in the pores of MOF). Subsequently, the MOF was further modified with cubic silver nanoparticles to obtain $Ag$-$Re_n$-MOF, thus the photocatalytic activity of $CO_2$ conversion was significantly improved (Figure 3A, [81]). The PXRD (powder X-ray diffraction) patterns showed that the single crystal $Re_3$-MOF structure is preserved when different amount of Re TC is introduced into $Re_n$-MOF (Figure 3B, [81]). By studying the process of photocatalytic conversion of $CO_2$ by $Re_n$-MOF (Figure 3C, [81]), it was found that the catalytic activity of $Re_3$-MOF was the highest. In addition, under visible light irradiation, the activity of $AgRe_3$-MOF was five times higher than that of $Re_3$-MOF, and the conversion efficiency of $CO_2$ to CO was increased by seven times. This is mainly because MOF has large porosity and $CO_2$ adsorption capacity, which is conducive to the occurrence of catalytic reduction reaction. On the other hand, precious metals have a wide range of photo absorption and are easier to trap photogenerated electrons due to the lower Fermi levels. At the same time, their stability could be further improved due to the strong covalent bond between Re TC and MOF.

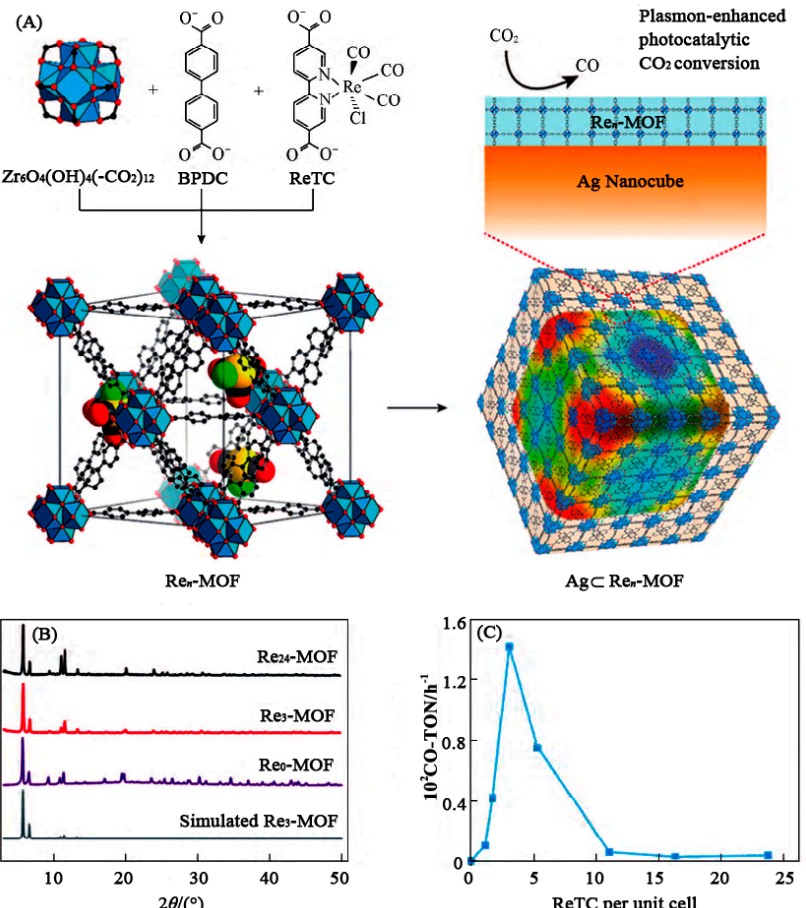

**Figure 3.** Structures of Ren-MOF and Ag Ren-MOF based catalysts (**A**), PXRD of Ren-MOFs (**B**), and the photocatalytic activity of Ren-MOF (**C**). Reprinted from ref. 81 with permission by the American Chemical Society.

Lee et al. [82] used UiO-66 (Zirconium 1,4-Carboxybenzene) as a precursor to obtain UiO-66-CAT with $Cr^{3+}$ or $Ga^{3+}$ sites as catalysts for photocatalytic $CO_2$ reduction. In the presence of TEOA and BNAH, the TON (turnover number) values of UiO-66-Cr CAT and UiO-66- Ga CAT are 11.22 ±0.37 and 6.14±0.22, and the amount of HCOOH that is produced by catalytic reduction of $CO_2$ after visible light irradiation for 6h were (51.73±2.64) and (28.78 ±2.52) μmol, respectively. The activity of UiO-66-Cr CAT is about twice higher than that of UiO-66-Ga CAT, which is mainly attributed to the fact that $Cr^{3+}$ is more efficient than $Ga^{3+}$ for the rapid transfer of electrons. At the same time, Cr derivatives show higher reduction efficiency than Ga derivatives due to their open shell structure.

Zhang et al. [83] reported Zr- porphyrin MOF (MOF-525-Co) as efficient catalysts for $CO_2$ conversion. Using TEOA as a sacrificial agent, MOF-525-Co could efficiently catalyze the reduction of $CO_2$ to CO and $CH_4$ under visible light irradiation. When compared with Zn-MOF-525 and MOF-525, MOF-525-Co showed the highest catalytic activity and $CO_2$ adsorption capacity. The metallized MOFs is obviously improved, and exhibited strong charge separation ability and energy conversion efficiency. The highest catalytic performance of cobalt metallized MOFs is mainly due to the fact that the introduction of monoatomic Co into MOF-525 can significantly improve the electron-hole separation efficiency in porphyrin ligands. At the same time, the photogenerated electrons rapidly migrated from the porphyrin center to the surface of the catalyst, thus the electrons with long lifetime were obtained, which effectively activated the $CO_2$ molecules that were adsorbed on the Co center.

Su et al. [84] prepared a series of $Cd_{0.2}Zn_{0.8}S$@UiO-66-$NH_2$ composites with different UiO-66-$NH_2$ content by solvothermal method, which were used for photocatalytic reduction of $CO_2$ to $CH_3OH$. The results showed that the single UiO-66-$NH_2$ showed no activity for photocatalytic $CO_2$ reduction,

but $Cd_xZn_{1-x}S$ with adjustable composition and band gap could be efficiently excited by visible light. All of the $Cd_{0.2}Zn_{0.8}S@UiO-66-NH_2$ samples showed excellent photocatalytic activity when compared with $Cd_{0.2}Zn_{0.8}S$. When the content of UiO-66-NH$_2$ was 20% (mass fraction), the catalyst showed the best photocatalytic activity, and the formation rate of $CH_3OH$ is 3.4 times higher than that of single structure $Cd_{0.2}Zn_{0.8}S$. This is mainly due to the effective charge separation and transfer at the interface between $Cd_{0.2}Zn_{0.8}S$ and UiO-66-NH$_2$. Thus, the photogenerated electrons that were absorbed by $Cd_{0.2}Zn_{0.8}S$ and UiO-66-NH$_2$ can be quickly transferred to the surface for $CO_2$ reduction. In addition, $Cd_{0.2}Zn_{0.8}S@UiO-66-NH_2$ photocatalyst showed excellent stability in the process of photocatalytic reduction of $CO_2$.

### 3.2. Zn MOFs

In 2015, Wang et al. [85] reported the establishment of $CO_2$ photoreduction system while using the CdS semiconductor and Co-ZIF-9 as catalyst and co-catalyst, respectively. Under mild reaction conditions, the reaction system of bipyridine and triethanolamine showed high catalytic activity when $CO_2$ was deoxidized to CO under visible light irradiation. Under the irradiation of monochromatic light at a wavelength of 420 nm, the quantum efficiency could reach 1.93%.

In 2018, Wang et al. [86] synthesized a series of ZIF-67 nanocrystals with a different morphology by the solvent induction method. Taking the advantages of MOF, the capture of $CO_2$ was controlled by controlling its morphology, and their photocatalytic performance was further improved. In the same year, Chen [87] and co-workers fabricated the Ag-Co-ZIF-9 nanocomposited materials with different Ag loading by the photo deposition method to study the effect of Ag NPs on the reaction performance of Co-ZIF-9 in $CO_2$ photo reduction reaction. In this study, Co-ZIF-9, with a rod structure was obtained by the reflux method, and ultra-small Ag nanoparticles (< 5 nm) were doped into Co-ZIF-9 by photodeposition. With the help of photosensitizer, the Ag@Co-ZIF-9 composite showed the catalytic performance of converting $CO_2$ to CO under the irradiation of visible light. With the increase of Ag nanoparticles, the formation of CO obviously increased while the amount of H$_2$ decreased. When compared with pure Co-ZIF-9, the photocatalytic activity of Ag@Co-ZIF-9 can be improved by two times (about 28.4 μmol CO), and selectivity about 20% (22.9 μmol H$_2$). The experimental results showed that Ag NPs in Co-ZIF-9 could act as an electron trap and active site for $CO_2$ reduction, thus the efficiency and selectivity of MOF materials in $CO_2$ photo reduction were improved.

Subsequently, Ye et al. [88] developed and used the ultra-thin two-dimensional Zn-MOF nanoliths to reduce $CO_2$ to CO. They firstly tried to establish two novel non-precious metal mixed photocatalytic systems. The catalyst showed excellent photocatalytic activity and selectivity under mild reaction conditions. It was confirmed that the Zn-MOF nanoparticles show better charge transfer ability than the Zn-MOF bulk materials via electrochemical impedance and PL (photoluminescence) spectroscopy analysis, thus stronger photocatalytic efficiency and selectivity were obtained. This provides feasibility for the application of photocatalysis in the development of various two-dimensional (2D) MOF materials.

In 2018, Zhao et al. [89] prepared $Zn_2GeO_4/Mg-MOF-74$ composites by the hydrothermal method (Figure 4). When the water was used as agent, the photocatalytic activity of $Zn_2GeO_4/Mg-MOF-74$ for $CO_2$ reduction reaction is higher than that of pure $Zn_2Ge_4$ nanorods or the physical mixture of $Zn_2GeO_4$ and Mg-MOF-74. This is mainly due to the stronger $CO_2$ adsorption performance of Mg-MOF-74, the lower recombination probability of photogenerated electron-hole pair and more alkali metal sites on the surface of Mg-MOF-74. In addition, the effect of H$_2$O on the reaction was also studied and the results show that H$_2$O is the reducing agent and hydrogen source involved in the reaction. In the process of reduction, the photogenerated electrons from the conduction band reduce $CO_2$ to CO and HCOOH, by the reaction of $CO_2+2e^-+2H^+\rightarrow HCOOH$ and $CO_2+2e^-+2H^+\rightarrow CO+H_2O$, in which the content of HCOOH is very small.

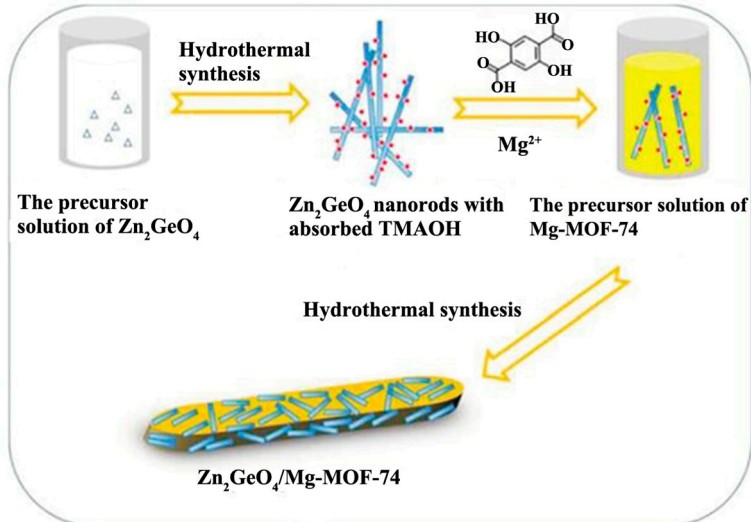

**Figure 4.** Schematic illustration of the synthesis of the $Zn_2GeO_4$/Mg-MOF-74 composites. Reprinted from ref. 89 with permission by the Royal Society of Chemistry.

In 2018, Cardoso et al. [90] modified $TiO_2$ nanotubes and formed a core-shell structure by layer growth of ZIF-8 nanoparticles on the surfaces. The FT-IR spectra show that the host-guest interaction depends on the pore structure and chemical properties of MOF connectors. Under UV irradiation at room temperature, $CO_2$ can be photocatalyzed to methanol and ethanol fuel on the electrode of composited materials. Zinc-based MOF not only provided the adsorption/activation of $CO_2$, but also acted as a light absorber to transfer excited electrons for photocatalytic reduction.

Sadeghi et al. [91] synthesized zinc-based porphyrin (Zn/PMOF), which could catalytically reduce $CO_2$ to $CH_4$ under light irradiation. The results showed that the yield of $CH_4$ was 10.43 μmol when Zn/PMOF was used as photocatalyst. After 4h irradiation, Zn/PMOF was much higher than that of $CH_4$ when ZnO was used as photocatalyst. At the same time, Zn/PMOF as photocatalyst showed high selectivity for $CO_2$ reduction, and it has better stability and repeatability when comparing to ZnO.

Yan et al. [92] loaded different amounts of $TiO_2$ on Co-ZIF-9 to construct Co-ZIF-9/$TiO_2$ nanostructure composites (ZIFx/T, x is the mass ratio of Co-ZIF-9 in the composites, T is $TiO_2$). The results showed that ZIF0.03/ T showed the best catalytic conversion efficiency of $CO_2$, and the yield of Ti/T is 2.1 times higher than that of pure $TiO_2$ catalyst after irradiation for 10h. Linear sweep voltammetry in $CO_2$ saturated solution further reveals that Co-ZIF-9 can effectively activate $CO_2$ and reduce the $CO_2$ reduction initiation potential of ZIFx/T (x ≤ 0.10). In addition, the photoluminescence spectra show that the ZIFx/T composites that were prepared by in-situ synthesis showed higher charge separation efficiency. Therefore, better $CO_2$ adsorption capacity and charge separation rate are beneficial to the high activity of ZIFx/T nanostructures in photocatalytic transformation.

Maina et al. [93] designed a catalytic system based on membrane reactor. The controllable encapsulation of $TiO_2$ and $Cu^{2+}$ doped $TiO_2$ nanoparticles (Cu-$TiO_2$) in ZIF-8 film was realized by the rapid thermal deposition (RTD) method (Figure 5A, [93]). Under ultraviolet irradiation, the Cu-$TiO_2$/ZIF-8 hybrid film showed high photocatalytic activity. The results show that, when compared with the amount produced by the original ZIF-8 film alone, the yields of CO and $CH_3OH$ increased by 188% and 50%, respectively (Figure 5B, [93]). Further studies showed that the yields of photocatalytic reduction of $CO_2$ to $CH_3OH$ and CO depend on the content of Cu-$TiO_2$ nanoparticles that are loaded on MOF films (Figure 5C, [93]). When the loading of Cu-$TiO_2$ nanoparticles is 7 μg, Cu-$TiO_2$/ZIF-8 exhibited the best catalytic efficiency. When compared with the original ZIF-8 film, the yields of CO and $CH_3OH$ increased by 23.3% and 70%, respectively. The sharp increase of product originated from the synergistic effect between the ability of semiconductor nanoparticles to produce photoexcited electrons under light irradiation and the high $CO_2$ adsorption capacity of MOF.

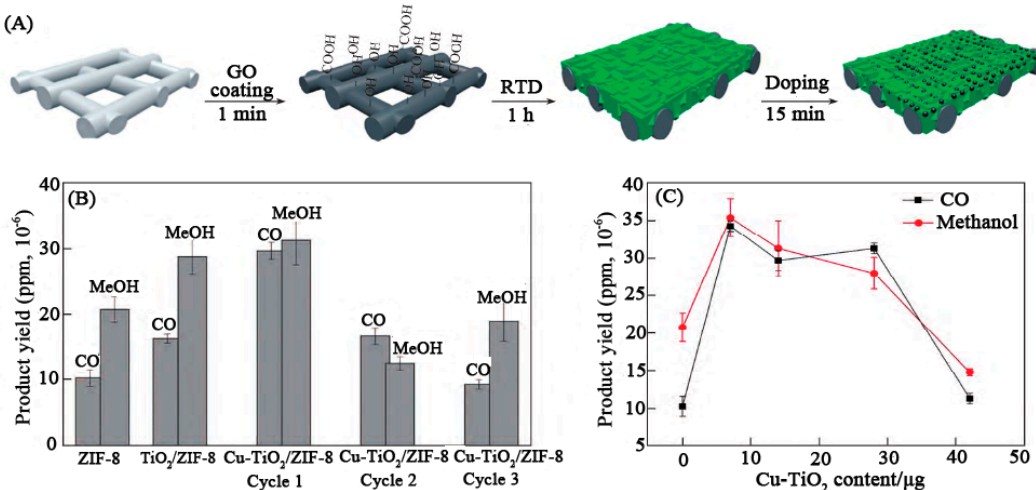

**Figure 5.** Fabrication of Cu-TiO₂/ ZIF-8 membranes (**A**), effect of membrane composition (**B**) and Cu-TiO₂nanoparticles loading on the product yields (**C**). Reprinted from ref. 93 with permission by the American Chemical Society.

Kong et al. [94] prepared CsPbBr₃@ ZIFs composites by in-situ synthesis used as $CO_2$ reduction photocatalyst with reinforcing activity (Figure 6A, [94]). The electron consumption rates of CsPbBr₃@ZIF-8 and CsPbBr₃@ZIF-67 are 15.498 and 29.630 $\mu mol \cdot g^{-1} \cdot h^{-1}$, which is 1.39 and 2.66 times higher than that of pure CsPbBr₃, respectively. The comparison of photocatalytic $CO_2$ reduction performance using CsPbBr₃ and CsPbBr₃@ZIFs showed that the ZIF coating greatly improved the catalytic activity of CsPbBr₃ (Figure 6B, [94]). In addition, six cycle experiments have been carried out on CsPbBr₃@ZIF, and it was found that the electron consumption rate suffered from negligible decrease. This indicates that it possessed good stability (Figure 6C, [94]). The synergistic effect of CsPbBr₃ and ZIF coating improved the stability of CsPbBr₃ to water molecules and enhanced the $CO_2$ capture ability and the charge separation efficiency. All of these lead to a higher conversion efficiency. Moreover, the catalytic active center Co in ZIF-67 could further accelerate the process of charge separation, activate $CO_2$ molecules, and improve the catalytic activity of $CO_2$ reduction.

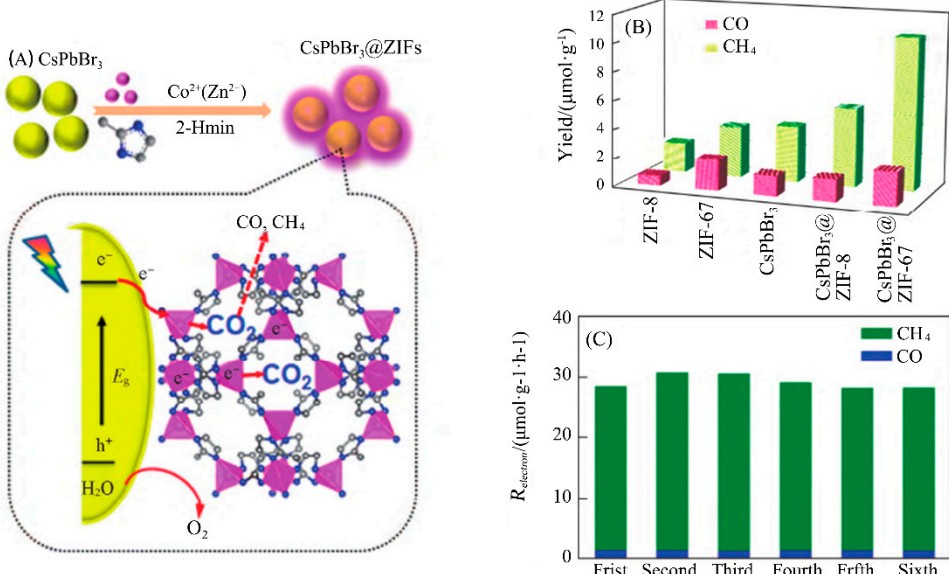

**Figure 6.** Schematic illustration of the fabrication process and $CO_2$ photoreduction process of CsPbBr₃/ ZIFs (**A**) and photocatalytic $CO_2$ reduction performances of CsPbBr₃ and CsPbBr₃@ZIFs (**B**,**C**). Reprinted from ref. 94 with permission by the American Chemical Society.

### 3.3. Ti MOFs

In 2012, Fu et al. [95] reported a photosensitive MOF $Ti_8O_8(OH)_4(bdc)_6(MIL-125(Ti))$ for photocatalytic $CO_2$ reduction. The photocatalytic activity evaluation indicated that Ti-MOF could efficiently reduce $CO_2$ to $HCOO^-$ under 365 nm UV irradiation. When comparing to other MOFs, MIL-125(Ti) showed slightly higher activity. The photocatalytic results of $NH_2$-MIL-125 showed that the concentration of $HCOO^-$ increased in the reaction system with the extension of irradiation time, and the formation of HCOO- reached 8.14 μmol within 10 hours. On one hand, the introduction of $NH_2$ can promote the rapid transfer of electrons from O to Ti, in $TiO_5(OH)$ metal cluster. On the other hand, $NH_2$ can significantly improve the adsorption capacity of $NH_2$-MIL-125 (Ti) to $CO_2$, which is beneficial for the adsorption and activation of $CO_2$ in the process of photocatalytic reaction. In 2018, He [96] designed an MOF-based ternal-composite photocatalyst $(TiO_2/Cu_2O/Cu_3(BTC)_2)$ to increase the density of charge carrier and promote the activation of $CO_2$ molecules to improve the photoreduction capacity of $CO_2$. The experimental results showed that the addition of $Cu_2O$ and $Cu_3(BTC)_2$ not only significantly improved the light conversion efficiency of $CO_2$, but also facilitated the formation of $CH_4$. The increase of charge carrier density improved the overall performance of the catalyst. The PL, XPS, and DRIFT analysis verified that the coordination of unsaturated metal sites were helpful in activating $CO_2$. This study provides a new way to solve the problems of low charge density and efficiency $CO_2$ activation, and it also provides a reasonable design for in-depth understanding of $CO_2$ photoreduction and other applications of mixed nanomaterials based on MOF.

### 4. Prospect of Photocatalytic CO$_2$ Reduction

The advantages of MOFs-based photocatalytic materials are obvious when comparing to conventional semiconductor materials. Thus, they have attracted more and more research attentions in photocatalysis. However, the low efficiency of this technology still hinders its wide applications in industry. The following problems should be addressed in the future. Firstly, researchers need to put forward effective strategies to improve the light absorption properties and charge separation performances. Secondly, most MOFs are not as metal oxide for semiconductor photocatalysts, especially in water or under ultraviolet light, which ultimately leads to the decreased catalyst life; hence, how to enhance their robustness is another important topic. Thirdly, there are few studies on the mechanism of photocatalytic $CO_2$ reduction in MOFs, especially the current understanding of the catalytic reaction path is still blurred. In addition, most of the reported photocatalytic $CO_2$ reduction reactions are carried out in organic solvents, requiring additional sacrificial agents. The future materials for catalytic reduction of $CO_2$ should be economical and environmentally friendly. Therefore, it is urgent to solve the above problems of MOFs materials for photocatalytic $CO_2$ reduction.

### 5. Conclusions

Artificial photosynthesis using catalysts to convert $CO_2$ to high value-added chemicals or fuels is an ideal way to effectively solve energy and environmental problems. The MOFs materials exhibit great application prospects in the field of photocatalysis, due to its ultra-high specific surface area, porous properties, modified/regulated textures, and high capture capability for $CO_2$ molecules. The advantages and significance of MOFs materials in $CO_2$ catalytic reduction are described in detail. Meanwhile, the application of typical MOFs in $CO_2$ photoreduction, for example, Zr-MOFs, Zn-MOFs, and Ti-MOFs, were introduced and summarized. Finally, the future development trend and prospect of photocatalytic $CO_2$ reduction are anticipated in this review.

**Funding:** This work was supported by the National Natural Science Foundation of China (No. 51572157), the Natural Science Foundation of Shandong Province (No. ZR2016BM16), Qilu Young Scholar Program of Shandong University (No. 31370088963043), and the Fundamental Research Funds of Shandong University (No. 2018JC036, 2018JC046).

**Conflicts of Interest:** The authors declare no conflict of interest. The funders had no role in the design of the study; in the collection, analyses, or interpretation of data; in the writing of the manuscript, or in the decision to publish the results.

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
