# Peer review of "Recent Advances in MOF-based Nanocatalysts for Photo-Promoted CO2 Reduction Applications"

_catalysts, doi:10.3390/catal9080658_

Round 1

Reviewer 1 Report

The paper is a brief review of MOF catalysts used for photo reduction of carbon dioxide to value added products. It seems to be quite comprehensive involving above 90 cited references. There are few question and remarks to be answered and discussed before publication:

i)          A table of content is missing in the beginning of the review.

ii)        English of the paper should be improved substantially. I suggest hiring a native English proof reader for this purpose.

iii)      Figures have to be improved as well. The pictures and letters are too small, resolutions are bad. Original papers the figure adopted from are not cited. It has to be complemented.

iv)      Unifying of catalytic results in a single table would be very informative. I know it is not easy task as different authors used different measures for characterizing activity. However, it would be task of the present authors to recalculate if it is possible all the data in order to have the same dimension for both activity and selectivity. In this point I have to mention that instead of TON if it is possible TOF should be used. TON is dimensionless value and dependent on the reaction time. In contrast to this TOF is a specific activity related to the reaction time and therefore it has a dimension of 1/time. TOF is not dependent on the reaction time.

v)        Mechanism of photoexcitation of MOFs should be described more precisely if the original papers touched this issue. It would be preferred to use schemes and figure in order to demonstrate energy levels and how the relaxation of energy of the photogenerated electron can occur over constituents of MOFs.

vi)      Page 3, line 127: “kinetic equilibrium” is a very strange expression. The thermodynamic equilibrium is a dynamic one where the forward and back reaction rate is equal to each other.

vii)    Page 4, line 155: What does “open metal site” mean? Perhaps authors wanted to write “accessible metal sites”.

After minor revision the paper can be published.

Author Response

Reply to Review comments

Journal: Catalysts

Manuscript ID: catalysts-553960

Title: Recent Advances in MOF-based Nanocatalysts for Photo-Promoted CO2 Reduction Applications

Authors: Chang Liu, Wenzhi Wang, Bin Liu, Jing Qiao, Longfei Lv, Xueping Gao, Xue Zhang, Dongmei Xu, Wei Liu, Jiurong Liu, Yanyan Jiang, Zhou Wang, Lili Wu and Fenglong Wang

Dear editor and Reviewers, 

We appreciate your constructive comments. We have responded to all the comments and addressed all the concerns in the revised manuscript. All the changes in the revised manuscript have been highlighted by giving the text a yellow background. We hope that our response will be able to satisfy the reviewer’s comments. Please see below the responses to reviewers:

Reviewers' comments:

The paper is a brief review of MOF catalysts used for photo reduction of carbon dioxide to value added products. It seems to be quite comprehensive involving above 90 cited references. There are few question and remarks to be answered and discussed before publication:

i)  A table of content is missing in the beginning of the review.

Reply: A table of content has been added in the beginning of the review (Page 5).

ii)  English of the paper should be improved substantially. I suggest hiring a native English proof reader for this purpose.

Reply: The English of manuscript has been improved in our whole paper.

iii)  Figures have to be improved as well. The pictures and letters are too small, resolutions are bad. Original papers the figure adopted from are not cited. It has to be complemented.

Reply: The Figures have been improved included pictures and letters. The related references have also been cited.  

iv)  Unifying of catalytic results in a single table would be very informative. I know it is not easy task as different authors used different measures for characterizing activity. However, it would be task of the present authors to recalculate if it is possible all the data in order to have the same dimension for both activity and selectivity. In this point I have to mention that instead of TON if it is possible TOF should be used. TON is dimensionless value and dependent on the reaction time. In contrast to this TOF is a specific activity related to the reaction time and therefore it has a dimension of 1/time. TOF is not dependent on the reaction time.

Reply: Thank for your good suggestion. Unfortunately, the reaction time values lacked in the current literature, thus the data are not enough to calculate TOF.

v)  Mechanism of photoexcitation of MOFs should be described more precisely if the original papers touched this issue. It would be preferred to use schemes and figure in order to demonstrate energy levels and how the relaxation of energy of the photogenerated electron can occur over constituents of MOFs.

Reply: Mechanism of photoexcitation of MOFs has been described more precisely (Page 4).

vi)  Page 3, line 127: “kinetic equilibrium” is a very strange expression. The thermodynamic equilibrium is a dynamic one where the forward and back reaction rate is equal to each other.

Reply: The strange expression “kinetic equilibrium” has been deleted in this paper.

vii) Page 4, line 155: What does “open metal site” mean? Perhaps authors wanted to write “accessible metal sites”.

Reply: The “open metal site” has been revised as “accessible metal sites” (Page 4, line 155).

Reviewer 2 Report

The review entitled “Recent Advances in MOF-based Nanocatalysts for Photo-Promoted CO2 Reduction Applications” have summarized the recent progress in this field and pointed out the current challenges of photocatalytic CO2 reduction using MOF-based materials. Additionally, the authors have described the unique advantages of metal-organic frameworks (MOFs)-based materials for photocatalytic reduction of CO2 and its capacity to solve the existing problems, and then the latest research progress in photocatalytic CO2 reduction has been recognized in detail. Moreover, the catalytic reaction process, conversion efficiency, as well as the product selectivity of photocatalytic CO2 reduction using MOFs based materials are thoroughly discussed. According to this review the present review has no novelty and no fruitful information to the researcher working in this research filed. Unfortunately, there are grammatical mistakes in English of the manuscript.  The authors are strongly requested to ask a native speaker to check English in the manuscript before submission in another journal. At present this review could not be accepted in "Catalysts"

Author Response

Reply to Review comments

Journal: Catalysts

Manuscript ID: catalysts-553960

Title: Recent Advances in MOF-based Nanocatalysts for Photo-Promoted CO2 Reduction Applications

Authors: Chang Liu, Wenzhi Wang, Bin Liu, Jing Qiao, Longfei Lv, Xueping Gao, Xue Zhang, Dongmei Xu, Wei Liu, Jiurong Liu, Yanyan Jiang, Zhou Wang, Lili Wu and Fenglong Wang

Dear editor and Reviewers, 

We appreciate your constructive comments. We have responded to all the comments and addressed all the concerns in the revised manuscript. All the changes in the revised manuscript have been highlighted by giving the text a yellow background. We hope that our response will be able to satisfy the reviewer’s comments. Please see below the responses to reviewers:

Reviewers' comments:

The review entitled “Recent Advances in MOF-based Nanocatalysts for Photo-Promoted CO2 Reduction Applications” have summarized the recent progress in this field and pointed out the current challenges of photocatalytic CO2 reduction using MOF-based materials. Additionally, the authors have described the unique advantages of metal-organic frameworks (MOFs)-based materials for photocatalytic reduction of CO2 and its capacity to solve the existing problems, and then the latest research progress in photocatalytic CO2 reduction has been recognized in detail. Moreover, the catalytic reaction process, conversion efficiency, as well as the product selectivity of photocatalytic CO2 reduction using MOFs based materials are thoroughly discussed. According to this review the present review has no novelty and no fruitful information to the researcher working in this research filed. Unfortunately, there are grammatical mistakes in English of the manuscript.  The authors are strongly requested to ask a native speaker to check English in the manuscript before submission in another journal. At present this review could not be accepted in "Catalysts"

Reply: The English of manuscript has been checked and corrected carefully in our whole paper.

Reviewer 3 Report

This paper is a review regarding MOF-based catalysts for CO2 reduction.  The authors show a lot of examples of MOF catalysts for the CO2 reduction in the text.  I think the paper is well organized and well written.  Therefore, this paper appears suitable for publication in this journal.  However, some minor things need to be corrected or considered as follows:

Comment 1: At line 106, a sentence, “this number will continuously growing”, needs to be corrected suitably.

Comment 2: At line 59, the activation of open-metal sites (removal of coordinated solvents) in HKKUST-1 and the application of open-metal sites should be briefly added.

e.g., Chem. Mater., 2017, 29, pp 26–39, DOI: 10.1021/acs.chemmater.6b02626

e.g., J. Am. Chem. Soc. 2019, 141, 7853- 7864, DOI: 10.1021/jacs.9b02114

e.g., Chem. Commun. 2018, 54, pp 6458-6471, DOI: 10.1039/C8CC02348D

Author Response

Reply to Review comments

Journal: Catalysts

Manuscript ID: catalysts-553960

Title: Recent Advances in MOF-based Nanocatalysts for Photo-Promoted CO2 Reduction Applications

Authors: Chang Liu, Wenzhi Wang, Bin Liu, Jing Qiao, Longfei Lv, Xueping Gao, Xue Zhang, Dongmei Xu, Wei Liu, Jiurong Liu, Yanyan Jiang, Zhou Wang, Lili Wu and Fenglong Wang

Dear editor and Reviewers, 

We appreciate your constructive comments. We have responded to all the comments and addressed all the concerns in the revised manuscript. All the changes in the revised manuscript have been highlighted by giving the text a yellow background. We hope that our response will be able to satisfy the reviewer’s comments. Please see below the responses to reviewers:

Reviewers' comments:

This paper is a review regarding MOF-based catalysts for CO2 reduction.  The authors show a lot of examples of MOF catalysts for the CO2 reduction in the text.  I think the paper is well organized and well written.  Therefore, this paper appears suitable for publication in this journal.  However, some minor things need to be corrected or considered as follows:

Comment 1: At line 106, a sentence, “this number will continuously growing”, needs to be corrected suitably.

Reply: The “this number will continuously growing” has been corrected as “this number will continuously grow”.

Comment 2: At line 59, the activation of open-metal sites (removal of coordinated solvents) in HKKUST-1 and the application of open-metal sites should be briefly added.

e.g., Chem.Mater., 2017, 29, pp 26–39, DOI:10.1021/acs.chemmater. 6b026 26 e.g., J. Am. Chem. Soc. 2019, 141, 7853- 7864, DOI: 10.1021/jacs.9b02114

e.g., Chem. Commun. 2018, 54, pp 6458-6471, DOI: 10.1039/C8CC02348D

Reply: The activation of open-metal sites (removal of coordinated solvents) in HKKUST-1 and the application of open-metal sites have been briefly added (Page 2). The related references have also been cited in this work. 

Round 2

Reviewer 2 Report

I have read carefully the revised review. The authors have incorporated all suggestions/corrections. Now this review could be accepted in its current format.